# Comparative Nanopore Sequencing-Based Evaluation of the Midgut Microbiota of the Summer Chafer (*Amphimallon solstitiale* L.) Associated with Possible Resistance to Entomopathogenic Nematodes

**DOI:** 10.3390/ijerph19063480

**Published:** 2022-03-15

**Authors:** Ewa Sajnaga, Marcin Skowronek, Agnieszka Kalwasińska, Waldemar Kazimierczak, Magdalena Lis, Monika Elżbieta Jach, Adrian Wiater

**Affiliations:** 1Laboratory of Biocontrol, Production, and Application of EPN, Centre for Interdisciplinary Research, The John Paul II Catholic University of Lublin, Konstantynów 1J, 20-708 Lublin, Poland; marcin.skowronek@kul.pl (M.S.); waldemar.kazimierczak@kul.pl (W.K.); magdalena.lis@kul.pl (M.L.); 2Department of Environmental Microbiology and Biotechnology, Nicolaus Copernicus University in Toruń, Lwowska 1, 87-100 Toruń, Poland; kala@umk.pl; 3Department of Molecular Biology, Institute of Biological Sciences, The John Paul II Catholic University of Lublin, Konstantynów 1J, 20-708 Lublin, Poland; monika.jach@kul.pl; 4Department of Industrial and Environmental Microbiology, Institute of Biological Sciences, Maria Curie-Skłodowska University, Akademicka 19, 20-033 Lublin, Poland; adrian.wiater@mail.umcs.pl

**Keywords:** gut microbiota, entomopathogenic nematodes, pathogen resistance, host–pathogen interactions, Scarabaeidae, nanopore sequencing, metataxonomics, pest biocontrol

## Abstract

Root-feeding *Amphimallon solstitiale* larvae and certain other scarab beetles are the main soil-dwelling pests found in Europe, while entomopathogenic nematodes (EPN) have been used as a biocontrol agent against these species. Our study provides the first detailed characterization of the bacterial community of the midgut in wild *A. solstitiale* larvae, based on the nanopore sequencing of the 16S rRNA gene. In the whole dataset, we detected 2586 different genera and 11,641 species, with only 83 diverse bacterial genera shared by all studied individuals, which may represent members of the core midgut microbiota of *A. solstitiale* larvae. Subsequently, we compared the midgut microbiota of EPN-resistant and T0 (prior to EPN exposure) individuals, hypothesizing that resistance to this parasitic infection may be linked to the altered gut community. Compared to the control, the resistant insect microbiota demonstrated lower Shannon and Evenness indices and significant differences in the community structure. Our studies confirmed that the gut microbiota alternation is associated with resistant insects; however, there are many processes involved that can affect the bacterial community. Further research on the role of gut microbiota in insect-parasitic nematode interaction may ultimately lead to the improvement of biological control strategies in insect pest management.

## 1. Introduction

The larvae of some members of the Scarabaeidae family (Coleoptera), such as *Amphimallon solstitiale*, the summer chafer, feed on roots and other organic matter. This can be extremely destructive to agricultural and forest ecosystems during population outbreaks [1,2]. These pests (also called white grubs) live underground in a complex soil environment and are difficult to control using pesticides, which can also affect the environment. Therefore, the development of alternative management strategies has received considerable interest, with a particularly promising use of entomopathogenic nematodes (EPN) [3,4]. However, the success of EPN as biological control agents depends on many abiotic and biotic factors. Unsuccessful host infection by EPN may also be a result of the evading behaviors, morphological barriers, and physiological traits of the insect host [5,6]. Although considerable research on the innate immunity of insect hosts has been carried out [7], little is known about the contribution of the insect microbiome to the course of EPN infection. Intestinal bacteria can influence host resistance against pathogens by contributing to immune system development, providing a range of essential nutrients and protection from pathogen colonization by competing for nutrients or producing inhibitory substances [8,9]. The knowledge of the interactions between gut residents, the insect host, and pathogens has great potential for the improvement of methods for managing harmful insects. Of particular interest is the prediction of the efficiency of control methods based on the composition of the insect pest microbiota or a strategy targeting the host’s microbiota for insect pest control [10,11].

Nematodes from the genera *Steinernema* and *Heterorhabditis* have engaged in mutualistic symbiosis with entomopathogenic *Xenorhabdus* and *Photorhabdus* bacteria, respectively (family *Morganellaceae*). The bacteria inhabit the gut of cognate nematode infective juveniles (IJs)—this is the infective stage of these parasites, which live in soil, seeking out and invading susceptible insect larvae. After gaining the insect hemocoel, IJs release their mutualistic bacteria, which replicate fast and spread over the insect body, killing the host rapidly due to septicemia [12]. It is known that the larval midgut is a site of penetration for EPN in the process of infection. The midgut microbiota appears to interact with EPN. However, it has not been elucidated whether the antagonistic bacteria that have been detected might protect the grubs from EPN, shaping resistance to this parasite pressure [6,13]. Although the information on the intricate bacterial community housed by the gut of scarab beetle larvae is available, research has mainly focused on the bacteria that are harbored in the hindgut and their contribution to the food digestion process [14,15]. To date, there are no comprehensive studies on the midgut bacterial community of scarab larvae.

Our previous study showed a shift in the bacterial composition in the midgut microbiome, associated with the resistance of the larvae of the common cockchafer, *Melolontha melolontha*, to EPN [16]. To verify these findings, we focused on the microbiota of another common scarab pest—the larvae of the summer chafer, *A. solstitiale*, the microbiota of which has not yet been studied in detail. Hypothesizing that differences in the composition of the host microbiota may be the cause of higher resistance to EPN or whether the resistance trait can modify the gut microbiota, we characterized the midgut bacterial microbiota of resistant individuals and compared them to those that were T0 (prior to EPN exposure). To avoid confounding the effects of the application of antibiotics on host physiology, which is an alternative and more commonly used method of testing the contribution of gut microbiota to resistance against pathogens, we studied the bacterial microbiota in resistant insects following exposure to a high dose of EPN. Additionally, we compared the gut microbiota of *A. solstitiale* larvae among different developmental stages and sampling sites. The application of the nanopore sequencing of the 16S rRNA gene, providing many advantages such as simple sample preparation and relatively long read lengths, ensured gut community profiling and allowed us to avoid the limitations of culture-dependent methods. This work highlights the importance of bacteria inhabiting the insect gut in insect–nematode interaction.

## 2. Materials and Methods

### 2.1. Collection and Preparation of Larvae

Second and third instar (L2 and L3) *A. solstitiale* larvae were sampled from two localities: a forest tree nursery and an urban lawn in the Lublin region, Eastern Poland (Table 1 and Appendix A (Appendix A)). The collection was performed in June 2019. The grubs were collected from the soil and placed separately in 50-mL plastic cups that were partly filled with soil taken from the sampling site. After transportation to the laboratory, the larvae were identified. In total, over 200 healthy *A. solstitiale* L2 and L3 larvae were selected for further study, following the protocol described previously in [16]. Shortly after, gut dissection was performed in 30 larvae, referred to as T0 (prior to EPN exposure). The extracted midguts were frozen and stored until processing. Simultaneously, the other larvae were placed in 1-L containers previously filled with soil and seeded with perennial ryegrass *Lolium perenne* L. (family Poaceae). The larvae were then exposed to *Heterorhabditis megidis*, *Steinernema arenarium*, *Steinernema bicornutum*, *Steinernema carpocapse*, and *Steinernema kraussei* at a dose of 2500 IJs per container. The experiment was carried out in controlled photoperiod and temperature conditions, according to the procedure described by Koppenhöfer and Fuzy (2004) [17]. We exposed 30 insect larvae to each EPN species (Appendix A, Appendix A). After 14 days of EPN treatment, the larvae were examined and qualified as EPN-resistant in the case of healthy-looking and active specimens, in contrast to the EPN-sensitive larvae, which were dead and with the characteristic cadaver turgor, color, and smell. The survival ratio was calculated as the proportion of viable larvae to dead ones after exposure to each EPN species.

### 2.2. DNA Extraction and Nanopore High-Throughput Sequencing

Total bacterial DNA was extracted from the *A. solstitiale* midguts using a Bead-Beat Micro AX Gravity kit (A&A Biotechnology, Gdynia, Poland). The obtained DNA concentration was monitored with Qubit 2.0 (Invitrogen, Carlsbad, CA, USA) using a Qubit dsDNA HS Assay Kit (Thermo Fisher Scientific, Waltham, MA, USA). Extraction of microbial community DNA, nanopore sequencing, data processing, and taxonomic identification were performed by genXone (Złotniki, Poland). For accurate detection of the bacterial communities, we targeted the fragment of the 16S rRNA bacterial gene spanning V3-V8 hypervariable domains, which were amplified with the forward primer F338 5′-ACTCCTACGGGAGGCAGC-3′ and the reverse primer R1391 5′-GACGGGCGGTGTGTRCA-3′ [18,19]. PCR amplicon libraries were prepared using a Ligation Sequencing Kit 1D and sequenced on a GridION X5 sequencer (Oxford Nanopore Technologies, Oxford, UK).

### 2.3. Bioinformatic Analysis

After the sequencing run, the reads were filtered for their quality using the following criteria: an average read quality ≥ 10 Phred, an average quality of read fragments with a window size of 500 bp ≥ 6 Phred, and a minimum sequence length of 800 bp. Adapters and barcodes were removed using Porechop v.0.2.4. High-quality reads were processed for taxonomic identification by matching the NGS sequences with sequences deposited in the NCBI RefSeq for the 16S ribosomal RNA database, using a modified BLAST algorithm. For each sample, an equivalent of the OTU frequency table was built using an Excel spreadsheet. Sequencing data were deposited in NCBI under the accession number PRJNA 787447.

### 2.4. Exploratory Data Analyses

To assess the sequencing depth, rarefaction curves were generated for all 28 samples that were sequenced, using phyloseq (v. 1.34.0) in R (v. 4.0.3). Rarefying to an even depth (82,308 reads) was performed with the same software package. The alpha diversity metrics comprised the observed number of species (Sobs), Shannon–Wiener biodiversity index (H’), Simpson biodiversity index (1/D), and community Evenness (E). These were calculated using the vegan (v. 2.5.7) package in R. Differences in the alpha diversity metrics between the group of samples that were subjected to the nematode pressure and the T0 group (prior to the EPN exposure) were tested with Student’s *t*-test or the Mann–Whitney test using Past v 3.08 [20]. Principal component analysis (PCoA) and an analysis of similarities (ANOSIM) between groups were performed using the vegan package in R. Additionally, to identify the differences without losing a considerable number of sequences due to the normalization procedure, the fold-changes in bacterial genera were calculated using DESeq2 in R [21]. Microbiome networks in the same groups were built using igraph, qgraph, vegan, and MCL in R [22]. They were made using the method of sparse correlations for compositional data (SparCC) as proposed by Friedman and Alm (2012) [23]. The link-analysis method was used to detect hubs or keystone taxa in the networks [24]. The keystone genera were nodes in the network that had a significantly larger number of links compared to the other hubs in the network. Keystone taxa were sorted according to the “hubbiness” scores, and the five top taxa from each group were presented and discussed further. Clusters were identified with the use of the “walktrap” algorithm [25].

## 3. Results

### 3.1. Resistance of A. solstitiale Larvae to EPN Infection

*A. solstitiale* larvae (L2 and L3 developmental stages) were collected from soils in eastern Poland. Selected larvae were assigned to the T0 or the EPN-resistant groups. Thirteen individuals, which constituted the T0 group, were not exposed to EPN, while fifteen randomly selected larvae, which constituted the EPN-resistant group, were selected after EPN exposure in the laboratory, based on the lowest level of susceptibility to this parasite (Table 1). Our survival test revealed that 52%, 68%, 66%, 57%, and 68% of the *A. solstitiale* larvae were healthy after 14 days of exposure to *S. arenarium*, *S. bicornutum*, *S. carpocapse*, *S. kraussei*, and *H. megidis*, respectively.

### 3.2. Nanopore Sequencing Results

The fourth-generation sequencing of the 16S rRNA gene using nanopore technology was applied to characterize the midgut bacterial community. The bioinformatics data processing resulted in 5,404,166 high-quality reads. The number of bacterial reads obtained across all the samples ranged from 82,317 to 646,404 (median 177,433). The rarefaction curves indicated a high coverage of bacterial diversity in the insect midguts; however, the curves for eight samples were still far from reaching the asymptote (Appendix A). The taxonomic classification rate varied from 86.7% to 99.8% (median 93.4%) at the genus level and from 69.3 to 93.2% (median 81.9%) at the species level (Table 1). In the whole datasheet, we identified 2519 different bacterial genera and 11,641 species in the range of 362–1746 and 1090–5324 for individual samples, respectively. However, since the base calling in the nanopore sequencing is still prone to errors, the results indicating the number of genera and species must be considered with caution. The bacterial genera present in all samples were considered as a core microbiome. This included 83 genera, constituting 3.3% of all identified genera. The most abundant were *Bacillus*, *Microbacterium*, *Paenibacillus*, *Turicibacter*, *Bacteroides*, *Anaerotruncus*, and *Bradyrhizobium*, whose average relative abundance (RA) exceeded 2%; however, a large inter-individual variation in the proportion of these genera was detected (Appendix A. The total overall relative abundance of all the core genera in the T0 and EPN-resistant groups of insects constituted 59.6% and 56.8% of the whole midgut community, respectively.

### 3.3. Bacterial Community Composition in the Midgut of the T0 Group of A. solstitiale Larvae

The analysis of the RA of the microbiota composition in the T0 group of larvae showed that the midgut was inhabited by bacteria belonging mainly to the phyla Firmicutes (48.9%), Proteobacteria (22.7%), Actinobacteria (14.6%), and Bacteroidetes (7.5%). Other detected phyla were present at RA < 1% (Figure 1, Appendix A). The phylum Firmicutes included mainly Clostridia, Bacilli, and Erysipelotrichia (28.8%, 11.3%, and 7.0%, respectively), while Proteobacteria were represented by α- and γ-Proteobacteria (11.2 and 7.8%, respectively). The class Clostridia was detected to have the highest RA in individual samples of the T0 group—up to 58.3%. The classes Actinobacteria and Bacteroidia were the widespread components in the T0 group samples as well, with RAs of 10.8% and 7.3%, respectively. Other detected classes displayed an RA of <2% (Appendix A, Appendix A).

The analysis of the T0 larval midgut community at the family level revealed 8 families: *Ruminococcaceae*, *Erysipelotrichaceae*, *Lachnospiraceae*, *Bacillaceae*, *Bacteroidaceae*, *Bradyrhizobiaceae*, *Propionibacteriaceae*, and *Hungateiclostridiaceae* with RA > 2% (Figure 1, Appendix A, Appendix A). The most abundant genera in the midgut of the T0 larvae were *Turicibacter*, *Bacteroides*, *Bacillus*, *Bradyrhizobium*, *Anaerotruncus*, and *Cutibacterium* (Appendix A), while *Turicibacter sanguinis*, *Anaerotruncus rubiinfantis*, and *Cutibacterium acnes* were the most abundant species (RA > 2%) (Appendix A, Appendix A).

**Figure 1 ijerph-19-03480-f001:**
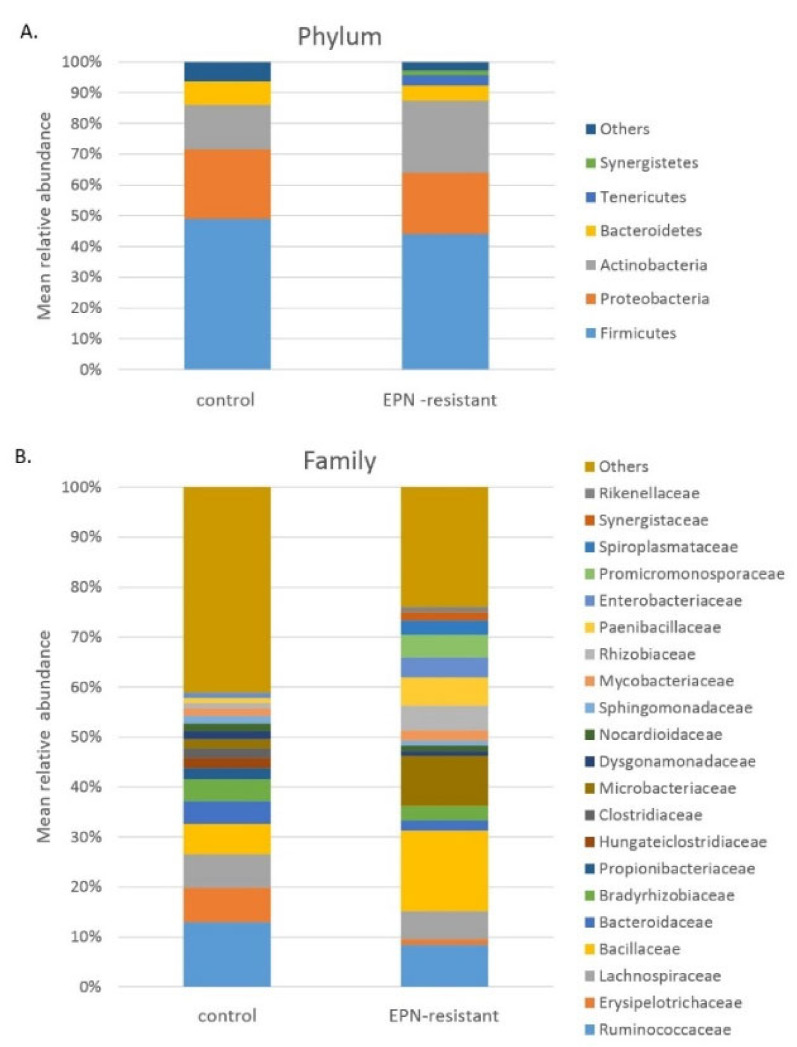
Composition of midgut bacteria of *A. solstitiale* larvae: (**A**) phylum level, (**B**) family level.

### 3.4. Bacterial Community Composition in the Midgut of the EPN-Resistant Group of A. solstitiale Larvae

We did not notice important differences between the microbiota of insects exposed to certain nematode species, either in the α-diversity indices or in the PCoA and ANOSIM analyses. Therefore, we included all the samples exposed to EPN to one group “EPN-resistant”. In the midgut of the EPN-resistant group, similar to the T0 group, Firmicutes (44.0%), Actinobacteria (23.5%), and Proteobacteria (20.0%) were the dominated phyla; however, Bacteroidetes, Tenericutes (also known as Mycoplasmatota), and Synergistetes were detected with relatively high abundance as well (5.0%, 3.4%, and 1.6%, respectively). Other phyla were present at RA < 1% (Figure 1, Appendix A). The most abundant classes of Firmicutes were Bacilli (23.4%) and Clostridia (18.2%), and the Bacilli class was detected with the highest RA in the individual samples of the EPN-resistant group—up to 87.1%. Other dominant phyla were Actinobacteria, α- and γ Proteobacteria, Bacteroidia, and Mollicutes (22.2%, 13.2%, 5.5%, 4.2%, and 3.4%, respectively). Other classes were detected with an RA < 2% (Appendix A, Appendix A).

The analysis of the midgut community of EPN-resistant larvae at the family level revealed more families, genera, and species, compared to the T0 group, with relatively high abundance. Families with an RA of >2% were represented by *Bacillaceae, Microbacteriaceae*, *Ruminococcaceae*, *Lachnospiraceae*, *Paenibacillaceae*, *Rhizobiaceae*, *Enterobacteriaceae*, *Promicromonosporaceae*, *Spiroplasmataceae*, *Bradyrhizobiaceae*, *Mycobacteriaceae*, *Mycobacteriaceae*, and *Bacteroidaceae* (Figure 1, Appendix A). The most abundant genera in the midgut of the EPN-resistant larvae were *Bacillus, Microbacterium, Paenibacillus, Spiroplasma, Enterobacter, Anaerotruncus, Rhizobium, Cellulosimicrobium, Virgibacillus*, and *Bacteroides* (RA > 2) (Appendix A). At the species level, the analysis revealed 4 species with an RA of >2%, i.e., *Bacillus pumilus, Spiroplasma ixodetis, Microbacterium kyungheense*, and *Anaerotruncus rubiinfantis* (Appendix A, Appendix A).

### 3.5. Comparison of the Midgut Microbiota between the T0 and the EPN-Resistant Groups of Insects

Although the mean value of the number of the observed genera in the T0 group of *A. solstitiale* was higher (1055) than in the EPN-resistant group (865), the difference was not significant (*p*-value 0.064). By contrast, both the community Evenness (E) and Shannon–Wiener (H’) diversity indices were higher in the T0 group of samples than in the group of EPN-resistant individuals (8.79 versus 5.71, *p*-value 0.009, and 4.36 versus 3.53, *p*-value 0.013, respectively). The median value of the Simpson diversity index (D-1) did not differ between the groups (8.21 and 6.29, *p*-value 0.059) (Table 2).

PCoA revealed a clear separation of 10 T0 samples from the rest of the specimens, indicating a different structure of the microbiota in the guts of insects subjected to the nematode exposure, prolonged incubation, and changed diet (Figure 2). The ANOSIM results fully supported this assumption (R 0.1781, *p*-value 0.006). The results of the same test performed on the T0 group only showed that the community structure in the midgut of the T0 group differed significantly, also given the location of the sampling site (R 0.409, *p*-value 0.016). However, the developmental stage did not exert such an effect (R 0.032, *p*-value 0.322).

The calculated fold changes revealed that the EPN-resistant larvae were enriched in 72 bacterial genera and were depleted in 96 genera. The most remarkable changes are presented in Figure 3. *Beduini, Stenotrophobacter, Neisseria*, unclassified *Rickettsieae, Pseudomaricurvus, Shewanella, Gemmata, Sunxiuqinia, Ignavibacterium*, and *Nordella* had significantly lower representation over the time of the experiment, i.e., comparing the group of the EPN-resistant specimens with the T0 group. On the other hand, *Cellulosimicrobium*, *Diaminobutyricibacter*, *Ochrobactrum*, *Laspinema*, *Ancylobacter*, *Xylanimicrobium*, *Rathayibacter*, *Plantibacter*, *Kaistia*, and *Propionicimonas* had much higher representation in the EPN-resistant larvae.

The microbiome networks that were created for the 100 most abundant genera of the two studied groups of *A. solstitiale* larvae differed substantially. The microbiome network of the EPN-resistant insects consisted of 3 clusters, while 2 clusters were detected in the T0 group (Table 3). The most important keystone genera were *Sphingomonas*, *Colidextribacter*, *Cloacibacillus*, *Bradyrhizobium*, and *Nocardioides* in the T0 group of larvae (Appendix A) and *Lachnoclostridium*, *Bosea*, *Mesorhizobium*, *Mycolicibacterium*, and *Devosia* in the EPN-resistant group (Appendix A, Appendix A*)*.

## 4. Discussion

### 4.1. Natural Midgut Microbiota of A. solstitiale Larvae

Several studies have shown that the digestive tract of scarab larvae contains a rich microbial community that may participate in various processes, especially digestion, and influence the immune system of the insect host [26]. However, little research has focused on the bacterial microbiota of the midgut, as this highly alkaline and oxidative compartment was considered to contain a less diverse bacterial community than in the hindgut, which is specialized for anaerobic fermentation [27,28]. In contrast, our data document a rich and diverse bacterial community associated with the midgut of *A. solstitiale* larvae (Coleoptera: Scarabaeidae). The considerable α-diversity indices of the midgut community were also observed in *M. melolontha*, *Melolontha hippocastani*, and *Pachnoda* spp., i.e., other Scarabaeidae members [16,29,30]. However, since different molecular techniques were used, a direct comparison of the α-diversity values seems to be improper.

In this study, the phyla Firmicutes, Proteobacteria, Actinobacteria, and Bacteroidetes were predominant in the analyzed midgut samples. The association of the midgut microbiota with mainly these four phyla has been reported for other rhizophagous larvae, e.g., *Melolontha* spp. [16,29]. However, while the phylum Proteobacteria dominated in the *M. hippocastani* midgut, Firmicutes representatives were dominant in *M. melolontha* and *A. *solstitiale**. This may probably be explained by the remote location of the site of *M. hippocastani* sampling, compared with the overlapping locations in the case of the latter ones. *Ruminococcaceae* and *Lachnospiracee* were the most abundant families in the midgut of *A. solstitiale* and *M. melolontha*, respectively, unlike *M. hippocastani*, which was dominated by *Enterobacteriaceae*. The latter bacterial family was also widespread in the whole gut of other wild coleopterans, such as *Nicrophorus vespiloides* and *Argilus planipennis* [31,32].

The developmental stage, diet, host environment, and host taxonomy were found to be key factors influencing the structure of gut bacterial communities [33,34]. Research demonstrated that the gut community of rhizophagous *M. hippocastani* larvae was more complex than the community of grazing adults; however, they had a subset of bacteria in common [29,32]. The differences in the microbiota of scarab larvae, observed at different instars fed an identical diet, were more discrete, and they seemed to increase proportionally to the larval age/developmental stage [29,35]. In our study, the gut bacterial communities from the L2 and L3 larvae were not different, which is in agreement with the results of our previous studies on the midgut microbiota of *M. melolontha* [16]; however, larva development-driven changes in the midgut microbiota composition might have been obscured by the high individual variation observed. On the other hand, we noticed a difference in the community structure of the *A. solstitiale* larvae between the sampling locations, which is probably linked with the diet. This agrees with earlier studies describing the bacterial gut community structure in scarab beetles, which was strictly influenced by diet; however, it was clearly distinct from the microbiota of ingested food [29,36,37]. Additionally, urbanization as an environmental factor may exert an impact on the microbial-insect host association in the soil [38,39].

The assumption that the diet and the environment habitat are the determinants of gut microbial assemblages in the root-feeding scarab larvae studied can be also supported by the large intraspecies differences observed in the gut microbiota composition. In our study, the difference in the gut bacterial composition between the samples was apparent, especially at lower taxonomic levels. In many beetles, gut bacterial communities vary among individuals within a species and appear to consist largely of bacteria that are not specifically adapted to living in the guts of their host species. Large intraspecific differences in the gut community structure were also reported for the larvae of other scarab beetles, such as the rhizophagous *M. melolontha* [16,28] and *Holotrichia parallela* [35], or the humivorous *Pachnoda* spp. [30].

The presence of both constant and variable components in the gut bacterial community has been shown in our study on *A. solstitiale* larvae. A total of 83 different bacterial genera, which constituted 3.3% of all genera detected, were common to all the midgut samples analyzed; however, this number may be underestimated, considering the possible inflation of diversity derived from nanopore sequencing data. Many genera found in the core bacterial microbiota of the *A. solstitiale* midgut were shared with those of *M. melolontha*, such as the most abundant *Turicibacter, Bacteroides*, and *Anaerotruncus*, among others [16]. This implies that these bacteria may play a key role in various digestive and protective processes in insects. For example, *Turicibacter sanguis*, abundantly detected in the insect gut microbiota, were formerly described as mainly fermentative bacteria, but the latest research suggests that it plays an important role in the promotion of insect fitness by linking with the host’s serotonergic system [40]. It has been suggested that the inconstant gut community in rhizogenous scarab beetle larvae that is reported in many insects, such as the *Pachysom**a* dung beetle or *Drosophila* spp., fits into the “minimal core” model initially described in mammals [36,41,42]. This model assumes weak partner fidelity and/or functional equivalence in diverse bacteria. On the other hand, we found a relatively high abundance of bacterial genera assigned to the core midgut microbiota of *A. solstitile* larvae (up to 60%), which may argue against favoring the minimal core microbiome concept. Nevertheless, since individual gut microbiota members can have an impact on host physiology, the ecological factors that determine natural variation in gut bacteria and the effect of this phenomenon on the fitness of their insect host are important issues for future study.

### 4.2. Alterations in the Gut Microbial Composition Associated with EPN-Resistance

It is known that EPN can infect a wide range of insects. However, this issue was frequently studied on lepidopteran caterpillars. A study on coleopteran *T. molitor* larvae confirmed their susceptibility to *Steinernema* spp. [43]. On the other hand, coleopteran larvae such as *A. solstitiale* or *M. melolontha* are relatively resistant to both *Heterorhabditis* and *Steinernema* nematodes [13,16,44]. This is consistent with the results of the survival assay, where we observed a high rate of larval survival following *Steinerema* and *Heterorhabditis* spp. treatment. Taking the impact of different EPN species exposure on the midgut microbiota community into account, we did not observe any significant differences. This is consistent with the findings reported by Cambon et al., who showed that the identity of three *Steinernema* spp. used for the infestation of insect larvae had no effect on the composition of the gut bacterial community in the early stage of infection [43]. However, it is also possible that we were not able to detect the discrete changes due to the high intra-individual variability of microbiota compositions.

Insects facing EPN associated with mutualistic bacteria co-evolved with them for millions of years, allowing the development of complex resistance mechanisms [5]. Several studies have reported that the composition of insect host microbiota is closely linked to the course of nematode parasitism [45,46]. Whether these resistance mechanisms, in addition to the activation of the immune response, avoidance behaviors, or morphological barriers, might include a selection of specific protective-acting microbes is still an open question. In addition to providing nutrients and improving the immune competence of the host against nematode infection, selection for resistance to nematodes may favor gut symbionts that act more directly, e.g., through the degradation of toxins released by nematodes and their native microbiota, the synthesis of a wide range of bioactive compounds that inhibit pathogen growth, or the alteration of the intestinal environment to limit successful EPN establishment. For example, certain bacterial endosymbionts of insects, *Wolbachia* and *Spiroplasma* spp., were found to be able to impede the growth of *Brugia* and *Howardula* spp. nematodes, which infect mosquitos and fruit flies, respectively [45,46]. Insect endosymbionts with antimicrobial activity, which are widespread in beetles, can also play an important role in the regulation of insect host–nematode interaction [47,48]. Previous studies showed that some members of the midgut microbiota of *A. solstitiale* and M. *melolontha* larvae, belonging to the *Pseudomonas, Serratia, Acinetobacter, Citrobacter, Bacillus, Enterococcu*, and *Paenibacillus* genera, are able to inhibit the growth of EPN symbiotic bacteria in vitro, implying that such activity can be one of the approaches to limit EPN infection in the natural world [13,44]. In all samples examined in this study, bacteria that are recognized as antagonistic to EPN symbionts were identified in the analyzed reads, reaching above the 0.001% RA thresholds (data not shown). This indicates that they are rare but widespread taxa, which is consistent with an equivalent study on *M. melolontha* [16].

The studied midgut bacterial communities showed considerable α-diversity in both analyzed groups of larvae; nevertheless, it was lower in the EPN-resistant group of larvae compared to the T0 group. These results are partly consistent with the study on *M. melolontha* larvae, where the α-diversity of midgut bacteria was comparably high, but we did not observe any significant difference between EPN-resistant and T0 individuals [16]. However, several studies on insects, including coleopteran soil-dwelling larvae, showed that resistance to entomopathogens is often correlated with simplified bacterial microbiota [49,50,51]. This finding may suggest that resistant *A. solstitiale* larvae living in the rich bacterial community of the soil are more selective in their internal filtering, which is aimed at avoiding species capable of becoming pathobionts in case of infection, thereby reducing the risk of secondary infections and septicemia, i.e., the major killing mechanism of EPN. Previous evidence has revealed that certain members of the insect gut community can facilitate *Steinernema* nematode pathogenesis, thus shaping the susceptibility of the host to infection in *Manduca sexta* and *Tenebrio molitor* larvae [43,52].

This study demonstrates that the composition of midgut bacteria is significantly different in the group of EPN-resistant insects, compared to the T0 group. Insect host resistance to parasites, bacteria, or toxins that are linked to an altered intestinal microbiota was also evidenced in several other studies [49,50,51,53,54]. Bacterial genera whose abundance is altered in *A. solstitiale* are usually rare members of the midgut community and belong to diverse phyla. Considering the most important changes, Actinobacteria were represented by a majority of genera that increased their abundance in the group with the EPN-resistant larvae compared to the T0 group, whereas the abundance of a majority of *Proteobacteria* decreased. In addition to the determination of differences in the composition of bacterial taxa in the microbiota, we assessed the keystone genera in both analyzed groups, which differed from each other. A similar study on *M. melolontha* bacterial microbiota also showed that the gut microbiota of the EPN-resistant group differed significantly in terms of the abundance of many genera, compared to the T0 group [16]; however, the different genera and the keystone genera differed from those shown in the present study. Nevertheless, the obtained results do not allow us to conclude whether the observed midgut bacterial microbiota alteration is a cause of resistance or rather a consequence of the EPN exposure.

In the PCoA analysis, the separation of the T0 group may rule out the explanation that specific bacteria were present in the resistant individuals before the exposure since, in this case, T0 individuals (which were not exposed to EPN) would likely be distributed among resistant ones. Therefore, the changes in the microbiota composition observed in the EPN-resistant larvae may be, at least partially, a consequence of EPN exposure, while the observed resistance of larvae to EPN is based on other mechanisms, probably a genetically determined immune response. This is in agreement with the study conducted by Tetreau et al., (2018), who concluded on the basis of a similar pattern of NMDS (Non-Metric Multidimensional Scaling) analysis that the modified microbiota associated with a high tolerance of *Bacillus thuringienis* was rather a consequence of the exposure of the host to this pathogen [50]. Similar to their study, our PCoA analysis displayed a higher dispersion among the resistant probes compared to the T0 group, supporting the concept of the EPN effect. In addition, the impact of the extended incubation and diet change in the case of the EPN-resistant individuals on the gut microbiota composition, leading to the spreading of these samples in PCoA analysis, could not be excluded. As previously shown, the gut microbiota of laboratory-reared insects fed with the simple diet is different (usually less diverse) from that in insects living in the wild [43].

On the other hand, in the PCoA analysis, we can observe three individuals from the T0 group that overlap with resistant insects, which suggests that they have changed microbiota and may represent the EPN-resistant phenotype, which is not connected with EPN exposure. A similar result was obtained in the experiments on the *M. melolontha* midgut [16]. In fact, the relationship between the gut microbiota the resistance to EPN is probably bidirectional. Selection for resistance to EPN can add pressure affecting the gut microbiota due to the activation of genes controlling bacterial overgrowth or the host’s immune response. Constantly increased immunity toward potential pathogens was detected in the *Bacillus thuringiensis*-resistant caterpillar *Galleria mellonella* [49].

Taken together, it is most probable that the changes observed in the midgut composition, apart from the extended incubation of larvae, are a result of the EPN treatment. It has been determined that EPN IJs highly transcribe genes coding for venom proteins already in the host-seeking stage, to be ready for their activation in the insect host; however, there are no data on the effects of these products on the host microbiota [55]. It is also known that other parasitic nematodes exhibit broad antimicrobial activity by secreting antibacterial factors such as lysozymes, cecropins, or C-type lectins; therefore, they can limit the growth of certain microbes and enrich the gut environment with nematode-protective microbiota members, supporting the suppression of host immunity and facilitating infection [45,56,57,58]. In addition, EPN mutualists are able to colonize the midgut, substantially affecting its microbiota, since they not only utilize gut nutrients but are also able to secrete a broad spectrum of antibiotics [59,60]. In turn, the role of non-mutualistic bacteria, isolated from different strains of *Steinernema* or *Heterorhabditis* nematodes, in their life cycle is unclear, but there is evidence that different secretively active bacteria can enter the insect gut with nematodes, such as *Alcaligenes* or *Paenibacillus* spp. [43,61].

Modification of the gut microbiome can also be an effect of tissue damage and changes in gut physiology, especially through the stimulation of the host’s innate immune system by the parasite, leading to increasing antimicrobial peptide production, which can keep gut microbiota under control [62,63]. Transcriptome analysis of *Holotrichia pararella* and *Drosophila melanogaster* treated with *Steinernema* and *Heterorhabditis* nematodes, respectively, indicated that, in the early stage of EPN infection, the activated genes were mostly associated with the immune system, especially the humoral response, leading to the secretion of antimicrobial peptides in gut epithelia [64,65]. Research has demonstrated that while Gram-positive bacteria are more sensitive to antibacterial defensins, antibacterial cecropins secreted into the gut lumen are active primarily against Gram-negative bacteria, such as *Xenorhabdus* and *Photorhabdus* bacteria [66,67,68]. This may support the suggestion that the observed decrease in the abundance of Proteobacteria is related to the insect immune response to the EPN mutualists.

On the other hand, direct toxin-mediated protection against the entomopathogens provided by defensive gut bacteria can not only mitigate EPN activity but also strongly affect the host and inhibit the members of its microbiota, particularly if the mechanisms are not specific, e.g., the production of bacteriocins or reactive oxygen species [9,69].

## 5. Conclusions

Despite the advent of increasing characterization of the insect pest microbiome, our understanding of the role of gut microbiota in determining infection outcomes is still insufficient. The application of more alternative tools to study the association between host microbiota and EPN calls for a determination of the role of specific gut residents, which is necessary for the elucidation of EPN resistance mechanisms. Our data confirm the high potential of 16S rRNA gene nanopore sequencing for profiling insect gut microbiota. Using this approach, we shed light on the midgut microbiota in wild coleopteran *A. solstitiale* larvae, highlighting compositional shifts in EPN infection-resistant individuals compared to T0 ones. Nevertheless, this work has some limitations, particularly with respect to the error-prone nanopore sequencing, inflating the biodiversity of the samples. Another limitation is the fact that we studied T0 samples (prior to the EPN exposure) as a point of reference; therefore, the observed effect can be biased by the lack of the same treatment of individuals of this group, compared to the EPN-resistant ones. Our study, however, is preliminary and we believe that the results are still relevant. Further studies on gut microbiota mediating the effect of host resistance to EPN are necessary, primarily to check whether insect resistance is dependent on nematode treatment. The high individual variation of the bacterial microbiota composition observed in this study may determine the possible effect of the microbiota on larval resistance to EPN, which is not uniform among individuals. Nevertheless, insects and their nematode parasites can be an excellent system for deciphering parasitic nematode strategies that result in pathogenic outcomes. Given the significance of scarab beetle pests, elucidation of insect-microbiota-pathogen interactions could potentially lead to the development of novel methods of pest biocontrol.

## Figures and Tables

**Figure 2 ijerph-19-03480-f002:**
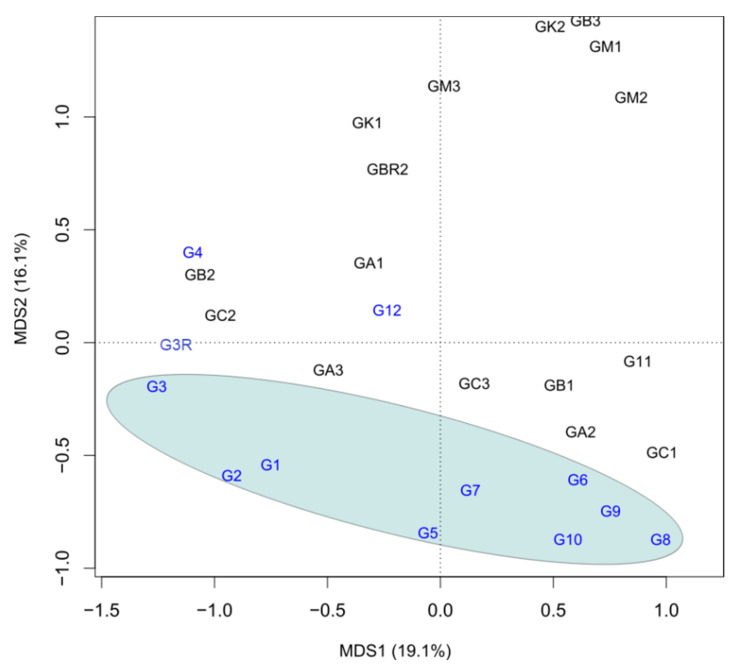
The principal coordinate analysis (PCoA) generated from the sequencing results of the 16S rRNA gene at the genus level. G—midgut samples of individuals from the T0 group (marked in blue); GA, GB, GC, GK, GM—midgut samples of individuals exposed to *S. arenarium*, *S. bicornutum*, *S. carpocapse*, *S. kraussei*, and *S. megidis*, respectively (marked in black). The numbers are an integral part of the sample name. All samples are further described in Table 1.

**Figure 3 ijerph-19-03480-f003:**
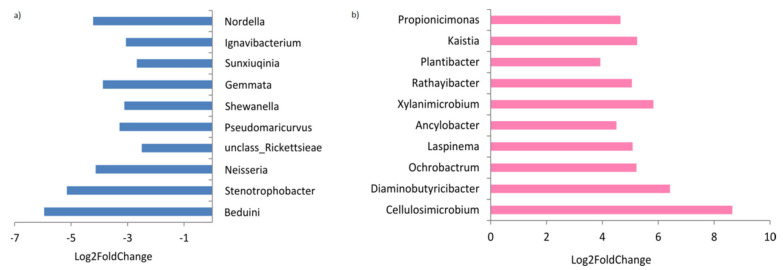
Average fold changes in the genera, with different abundances in the midgut of both the T0 and EPN-resistant groups of *A. solstitiale* larvae. (**a**) Negative changes, (**b**) positive changes (*p* < 0.05).

**Table 1 ijerph-19-03480-t001:** Sampling sites, NGS sequencing statistics, and α-diversity metrics for the studied midgut bacterial microbiota.

Sample Name	Nematode Exposure	Sampling Site *	Developmental Stage of Larva	Total Number of Reads	Classification Rate (%)		Diversity Metrics			
					Genus Level	Species Level	Sobs	Shannon	Simpson	Evenness
G1	No	FN	L2	252.226	91.8	79.1	890	4.63	0.98	0.68
G2	No	FN	L2	175.056	96.1	83.8	646	3.49	0.93	0.54
G3	No	FN	L3	191.586	93.7	86.1	967	4.38	0.97	0.64
G3R	No	FN	L3	130.810	89.1	85.7	914	4.09	0.95	0.60
G4	No	FN	L3	161.824	95.9	91.8	1157	4.56	0.97	0.65
G5	No	FN	L2	666.345	91.0	86.8	977	4.25	0.97	0.62
G6	No	FN	L2	202.974	86.7	81.8	969	3.96	0.91	0.58
G7	No	FN	L2	193.900	89.4	83.4	862	4.22	0.95	0.62
G8	No	UL	L3	139.289	95.1	81.5	1464	5.32	0.99	0.73
G9	No	UL	L3	83.109	93.8	87.7	1265	4.42	0.92	0.62
G10	No	UL	L3	112.619	93.0	75.3	1378	4.82	0.98	0.67
G11	No	UL	L3	194.582	90.7	79.1	1182	4.33	0.95	0.61
G12	No	UL	L3	97.439	93.4	83.7	1051	4.25	0.95	0.61
GA1	Yes	FN	L2	137.310	92.7	86.9	647	3.59	0.93	0.55
GA2	Yes	FN	L2	107.186	92.6	81.3	987	4.60	0.98	0.67
GA3	Yes	FN	L2	173119	97.6	93.2	622	2.85	0.79	0.44
GB1	Yes	FN	L2	146.971	93.1	78.2	1360	5.01	0.98	0.69
GB2	Yes	FN	L2	181.119	93.3	86.7	931	3.93	0.95	0.58
GB2R	Yes	FN	L2	199.582	96.0	77.0	225	0.83	0.28	0.15
GB3	Yes	FN	L2	217.014	99.7	69.3	926	3.26	0.87	0.48
GC1	Yes	FN	L2	216.091	93.2	78.1	1161	4.89	0.98	0.69
GC2	Yes	FN	L2	175.533	93.2	85.8	851	4.04	0.96	0.60
GC3	Yes	FN	L2	217.951	92.1	81.9	1228	3.35	0.79	0.47
GK1	Yes	FN	L2	158.666	95.9	74.8	521	2.37	0.76	0.38
GK2	Yes	FN	L2	257.766	96.3	74.6	848	3.37	0.88	0.50
GM1	Yes	FN	L2	129.209	96.8	73.3	797	3.33	0.91	0.50
GM2	Yes	FN	L2	300.879	95.1	81.3	898	3.71	0.94	0.55
GM3	Yes	FN	L2	244.162	94.7	85.0	970	3.87	0.93	0.56

* FN: forest nursery at 51°23′37.0″ N 22°29′44.0″ E; UL: urban lawn at 51°14′19.7″ N 22°29′58.7″ E.

**Table 2 ijerph-19-03480-t002:** Comparison of diversity metrics between the T0 group of *A. solstitiale* larvae and EPN-resistant larvae.

Diversity Metrics	T0	EPN-Resistant	Statistic	Statistic-Value	*p*-Value
	N = 13	N = 15			
Observed	1055	865	t	1.94	0.064
Shannon	4.36	3.53	t	2.82	0.013
Simpson	8.21	6.29	Mann–Whitney U	56	0.059
Evenness	8.79	5.71	Mann–Whitney U	40	0.009

**Table 3 ijerph-19-03480-t003:** Basic characteristics of microbiome networks in the T0 group of *A. solstitiale* larvae and EPN-resistant larvae.

	T0 Group	EPN-Resistant Group
	(N = 13)	(N = 15)
Keystone genera	*Sphingomonas* (1.00 *)	*Lachnoclostridium* (1.0)
	*Colidextribacter* (0.99)	*Bosea* (0.99)
	*Cloacibacillus* (0.98)	*Mesorhizobium* (0.98)
	*Bradyrhizobium* (0.96)	*Mycolicibacterium* (0.96)
	*Nocardioides* (0.95)	*Devosia*
Number of clusters	2	3
Modularity	0.067	0.126

* hubbiness scores are given in parentheses.

## Data Availability

The authors confirm that the data supporting the findings of this study are available within the article and its Appendix A. Raw data that support the findings have been deposited in Sequenced Read Archive (SRA) under accession number PRJNA 787447.

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
