# Peer review of "Comparative Nanopore Sequencing-Based Evaluation of the Midgut Microbiota of the Summer Chafer (Amphimallon solstitiale L.) Associated with Possible Resistance to Entomopathogenic Nematodes"

_ijerph, 2022, doi:10.3390/ijerph19063480_

Round 1
Reviewer 1 Report
In this study, Sajnaga et al. have employed nanopore sequencing of the 16S rRNA 20 gene to characterize the midgut microflora of EPN-resistant and non-resistant Amphimallon solstitiale. Based on the fact that Amphimallon solstitiale are pests difficult to control with traditional pesticides, this research not only showcases the potential of using nematodes to manage outbreaks but also unravels the importance of gut microbiota in gaining resistance to such nematodes.
In this sense, the study is sound, designed and executed in an appropriate manner. The data are novel and well-discussed. I have some small suggestions to be considered and/or incorporated:
- among the vast set of data, the authors could point our to the most promising Phyla/Family or species that could play an important role in the acquisition of resistance and ellaborate a few ideas/hypotheses in the discussion.
- what could be the limitations of the study?
- Figure 1 is rather small and difficult to read. i would suggest to increase its dimensions or resolution
- the authors shoul re-check the refernces listed in the main body of the manuscript since a handfull is not referenced according to the requirements of the journal.
Author Response
List of responses - REVIEWER 1
We would like to thank the reviewers for the useful comments to improve the paper. We have addressed all the comments as explained below.
- The authors should point out to the most promising Phyla/Family or species that could play an important role in the acquisition of resistance and elaborate a few ideas/hypotheses in the discussion.
According to the Reviewer’s remark, the information on bacterial taxa that could be an important player in conferring resistance against entomopathogenic nematodes has been added to the discussion:
“For example, bacterial endosymbionts of insects, Wolbachia and Spiroplasma spp., were found to be able to impede the growth of Brugia and Howardula spp. nematodes, which infect mosquitos and fruit flies, respectively [45, 46]”
Additionally, some sentences have been changed to highlight the possible defensive role of some insect endosymbionts against parasitic nematodes: “Insect endosymbionts with antimicrobial activity, which are widespread in beetles, can also play an important role in regulation of the insect host–nematode interaction [47, 48]. Previous studies showed that some members of the midgut microbiota of A. solstitiale and M. melolontha larvae, belonging to Pseudomonas, Serratia, Acinetobacter, Citrobacter, Bacillus, Enterococcu, and Paenibacillus genera, are able to inhibit the growth …”
- What could be the limitations of the study?
According to the Reviewer’s remark, the description of study limitations has been added to the Conclusion subsections:
“Nevertheless, this work has some limitations, particularly with respect to the error-prone nanopore sequencing, inflating the biodiversity of the samples. Another limitation is the fact that we studied T0 samples (prior to the EPN exposure) as a point of reference; therefore, the observed effect can be biased by the lack of the same treatment of individuals of this group comparing to the EPN-resistant ones. Our study, however, is preliminary and we believe that the results are still relevant”
- Figure 1 is rather small and difficult to read. I would suggest to increase its dimensions or resolution
The dimension of Figure 1 has been increased.
- The authors should re-check the references listed in the main body of the manuscript since a handful is not referenced according to the requirements of the journal.
The reference list has been checked thoroughly and changes have been made in position 5, 6, 10, 12, 14, 58, and 69, according to the requirement of the journal.

Reviewer 2 Report
Sajnaga et al. presents partial-16S nanopore data of the bacterial communities in the midgut of A. solstitiale (Scarabaeidae) before and after exposure to parasitic nematodes. The communities are compared between larvae coming from 2 different locations, 2 different developmental stages and after exposure to one of 5 different nematodes. Most emphasis is put on the comparison of the communities in the gut before and after exposure to the nematodes. The study is well-written, timely and provides novel insights. As such, I think it will be of interest to the reads of ijerph.
I have two major remarks which I think should be addressed prior to acceptance: (1) the interpretation of the ‘control’ treatment and (2) the interpretation of nanopore data derived diversity estimates.
The control treatment is defined as the individuals prior to the exposure to the nematodes and compared to those after exposure to the nematodes. It is therefore not a ‘control’, which would be the individuals subject to the same treatment (placed in 1-liter containers filled with soil and seeded with L. perenne and incubated for 14 days) where nematodes were omitted, but a ‘before’ or ‘T0’. This is an important distinction as the comparison between T0 and the nematode exposed samples will also reflect the effects that the incubation, and feeding on L. perenne will have on the gut microbiome. Below I highlight several points in the manuscript that should be adjusted to reflect this better.
Since the basecalling in the nanopore sequencing is still prone to errors, there will be a significant amount of basecalls that will be wrong (as reflected by the low Phred scores used for filtering). In Illumina, which produces much higher Phred scores, this is further complemented by a clustering or read correcting approach. In the absence of any advanced read corrections or a read consensus approach, BLAST results of individual reads will often be wrong and inflate diversity and the number of observed genera/species (identification on lower taxonomic levels, e.g. family, is likely to be more robust). The overestimation of diversity is also noticeable in the rarefaction curves that are not saturating despite the high number of reads/sample. Comparison of species richness between samples remains valid as the samples within this study all experience the same inflation problem. However, absolute quantifications (e.g. number of genera observed) and comparisons to other studies are not valid. Below I highlight several points in the manuscript that should be adjusted to reflect this better.
Minor comments:
L56-57: please clarify
L60-61: would remove ‘however and phrase as two sentences
L78-79: since you are not sequencing the entire 16S and not correcting for sequencing errors, species identification will be unreliable. You can refer to other nanopore benefits, fast turnover times, in-house sequencing possibility … if you want to emphasis novelty of this study with regard to nanopore sequencing
L92: ‘control’ -> ‘T0’ or ‘prior to exposure’; please also adjust later in the manuscript
L94: it might be nice for some readers to know that Lolium is a Poaceae/grass
L125: it could be interesting to mention that, in total 28 samples were sequenced
L126;134: please provide the versions of R, phyloseq and vegan
L137-139: please clarify how you define ‘keystone taxa’ from these networks
L155-156: most of them are still steeply going up
L160: these number are likely to be overestimations (by multiple factors)
L164: can you also provide an indicate of overall relative abundance of these general, is this group generally highly abundant across samples?
L192-213: where there any important differences between different nematode species?
L226-228: a different structure after the experiment compare to before (nematode+ incubation in the container=feeding on a single grass species)
L238-245: these represent increased or decreased abundance over time not control vs exposed
L251: have you tested for differences between networks. if not, please replace ‘significantly’ with ‘substantially’
L251-259: should clades not be clusters? These are not visible in the supp figure
L267-272: this comparison is nearly impossible to make as different sequencing technologies and pipelines were used so I would phrase it more carefully, putting more emphasis on this issue.
L322: I don’t know if it is so minimal, considering the inflation of diversity. What happens if you look at the overall relative abundance of these core taxa?
L356: likely not such a high alpha diversity
L385-396; L407-409: please rephrase to reflect that time/incubation conditions might have also played an important role
L396: what about differences between nematode species, not really discussed in this section?
L442: advanced -> alternative
L450: independent -> dependent
Author Response
List of responses - REVIEWER 2
We would like to thank the reviewers for the useful comments to improve the paper. We have addressed all the comments as explained below.
Major remarks:
- the interpretation of the ‘control’ treatment
- the interpretation of nanopore data derived diversity estimates
These remarks were very valuable to us because we are, in fact, beginners in the field of relationships between insect gut microbiota and entomopathogenic nematodes. We have added the issues of 1/ the lack of true control samples and, instead, including T0 (prior to the EPN infection) samples as the point of reference and 2/ the overestimation of biodiversity in analysis based solely on nanopore sequencing to the manuscript. They have been described as the main limitations of our study in the Conclusion subsection:
“Nevertheless, this work has some limitations, particularly with respect to the error-prone nanopore sequencing, inflating the biodiversity of the samples. Another limitation is the fact that we only studied T0 samples (prior to the EPN exposure) as a point of reference; therefore, the observed effect can be biased by lack of the same treatment of individuals of this group comparing to the EPN-resistant ones. Our study, however, is preliminary and we believe that the results are still relevant”.
Minor comments:
L56-57: please clarify
One sentence has been added and one changed for better clarification of the EPN bacteria life cycle: “The bacteria inhabit the gut of cognate nematode infective juveniles (IJs) - infective stage of these parasites, which live in soil, seeking and invading susceptible insect larvae. After gaining the insect hemocoel, IJs release their mutualistic bacteria, which replicate fast and spread over the insect body, killing the host rapidly due to septicemia [12].”
L60-61: would remove ‘however and phrase as two sentences
This phrase has been split into 2 separate sentences: “The midgut microbiota appears to interact with EPN. However, it has not been elucidated whether the antagonistic bacteria detected might protect grubs from EPN, shaping resistance to this parasite pressure [6,13]”
L78-79: since you are not sequencing the entire 16S and not correcting for sequencing errors, species identification will be unreliable. You can refer to other nanopore benefits, fast turnover times, in-house sequencing possibility … if you want to emphasis novelty of this study with regard to nanopore sequencing
This has been corrected according to the Reviewer’s suggestion: “The application of the nanopore sequencing of the 16S rRNA gene, providing many advantages such us simple sample preparation and relatively long read lengths, ensured gut community profiling and allowed avoiding the limitation of culture-dependent methods.”
L92: ‘control’ -> ‘T0’ or ‘prior to exposure’; please also adjust later in the manuscript
This has been corrected: “Shortly, gut dissection was performed in 30 larvae, referred to as T0 (prior to the EPN exposure).”
Additionally, the term “T0 group” has been introduced in the manuscript and supplementary data instead of “control”.
L94: it might be nice for some readers to know that Lolium is a Poaceae/grass
This has been corrected: “…the other larvae were placed in 1-liter containers previously filled with soil and seeded with Lolium perenne L. grass (family Poaceae)”
L125: it could be interesting to mention that, in total 28 samples were sequenced
To assess the sequencing depth, rarefaction curves were generated for all 28 samples sequenced, using phyloseq (v. 1.34.0) in R (v. 4.0.3).
L126;134: please provide the versions of R, phyloseq and vegan
Versions of R, phyloseq and vegan, have been added accordingly:
“To assess the sequencing depth, rarefaction curves were generated using phyloseq (v. 1.34.0) in R (v. 4.0.3). Rarefying to even depth (82,308 reads) was done with the same software package. The alpha diversity metrics: the observed number of species (Sobs), Shannon-Wiener biodiversity index (H’), Simpson biodiversity index (1/D), and community Evenness (E) were calculated using vegan (v. 2.5.7) in R.”
L137-139: please clarify how you define ‘keystone taxa’ from these networks
We have clarified the definition of the keystone taxa: “The link-analysis method was used to detect hubs or keystone taxa in the networks [24]. Keystone genera were nodes in the network that had a significantly larger number of links compared to the other hubs in the network. Keystone taxa were sorted according to the hubbiness scores and five top from each group were presented and discussed further.”
L155-156: most of them are still steeply going up
We have added a proper comment: The rarefraction curves indicated a high coverage of bacterial diversity in the insect midguts; however, the curves for eight samples were still far from reaching the asymptote (Figure S2).
L160: these number are likely to be overestimations (by multiple factors)
We have added a proper comment: “In the whole datasheet, we identified 2519 different bacterial genera and 11641 species in the range of 362-1746 and 1090-5324 for individual samples, respectively; however, since the basecalling in the nanopore sequencing is still prone to errors, the results of the number of genera and species must be considered with caution.
L164: can you also provide an indicate of overall relative abundance of these general, is this group generally highly abundant across samples?
The proportions of the most abundant genera for each sample are given in Table S2, while the relative abundance of these genera with a 95% confidence interval for each experimental group is given in Table S5. To highlight the inter-individual variation in the relative abundance of these genera, the phrase has been added: “The most abundant were Bacillus, Microbacterium, Paenibacillus, Turicibacter, Bacteroides, Anaerotruncus, and Bradyrhizobium, whose average relative abundance (RA) exceeded 2%; however, a large inter-individual variation in the proportion of these genera was detected (Table S2; Table S5).”
L192-213: where there any important differences between different nematode species?
We have added proper information: “We did not notice important differences between the microbiota of insects exposed to certain nematode species either in the α-diversity indices or in the PCoA and ANOSIM analysis.”
L226-228: a different structure after the experiment compare to before (nematode+ incubation in the container=feeding on a single grass species)
We have changed it accordingly: “PCoA revealed a clear separation of 10 T0 samples from the rest of the specimens, indicating a different structure of the microbiota in the guts of insects subjected to the nematode exposure, prolonged incubation, and changed diet (Figure 2)”
L238-245: these represent increased or decreased abundance over time not control vs exposed
We have changed it accordingly: “The most remarkable changes are presented in Figure 3. Beduini, Stenotrophobacter, Neisseria, unclass Rickettsieae, Pseudomaricurvus, Shewanella, Gemmata, Sunxiuqinia, Ignavibacterium, and Nordella had significantly lower representation over the time of the experiment, i.e. comparing the group of the EPN-resistant with T0.”
L251: have you tested for differences between networks. if not, please replace ‘significantly’ with ‘substantially’
We have it changed accordingly: “The microbiome networks created for the hundred most abundant genera of the two studied groups of A. solstitiale larvae differed substantially”.
L251-259: should clades not be clusters? These are not visible in the supp figure
We have changed it accordingly: “The microbiome networks created for the hundred most abundant genera of the two studied groups of A. solstitiale larvae differed substantially. The microbiome network of the EPN-resistant insects consisted of 3 clusters, while 2 clusters were detected in the control group (Table 3)”.
L267-272: this comparison is nearly impossible to make as different sequencing technologies and pipelines were used so I would phrase it more carefully, putting more emphasis on this issue.
According to the Reviewer’s suggestion, the language of this part has been softened and direct α diversity comparison has been removed: “The considerable α-diversity indices of the midgut community were also observed in M. melolontha Melolontha hippocastani, and Pachnoda spp., i.e. other Scarabaeidae members [16, 29,30]. However, since different molecular techniques were used, direct comparison of the α-diversity values seems to be improper”.
L322: I don’t know if it is so minimal, considering the inflation of diversity. What happens if you look at the overall relative abundance of these core taxa?
We have calculated that the total overall relative abundance of all the core genera in a given insect group constituted 59.6% and 56.8% of the whole midgut community, respectively. Indeed, in such a view, we cannot use the concept of the minimal core. Therefore, the part in the Discussion subsection has been corrected accordingly:
“A total of 83 different bacterial genera, which constituted 3.3% of all genera detected, were common to all the midgut samples analyzed; however, this number may be underestimated considering the possible inflation of diversity derived from nanopore sequencing data. Many genera found in the core bacterial microbiota of A. solstitiale midgut were shared with these of M. melolontha, such as the most abundant Turicibacter, Bacteroides, and Anaerotruncus among others [16]. (…) It has been suggested that the inconstant gut community in rhizogenous scarab larvae reported in many insects, such as the Pachysoma dung beetle or Drosophila spp., fits into the “minimal core” model, initially described in mammals [36, 41, 42]. It assumes weak partner fidelity and/or functional equivalence in diverse bacteria. On the other hand, we found relatively high abundance of bacterial genera assigned to core midgut microbiota of A. solstitile larvae (up to 60%), which may argue against favoring the minimal core microbiome concept.”
Additionally, for clarification of this issue, the following phrase has been added to the Results subsection: “The total overall relative abundance of all the core genera in the T0 and EPN-resistant group of insects constituted 59.6% and 56.8% of the whole midgut community, respectively”.
L356: likely not such a high alpha diversity
We have rephrased the sentence as follows: “The studied midgut bacterial communities showed considerable α-diversity in both analyzed groups of larvae (…)”
L385-396; L407-409: please rephrase to reflect that time/incubation conditions might have also played an important role
According to the suggestion, we have elaborated the issue of the impact of insect incubation on their gut microbiota composition in the Discussion subsection:
L385-396 “In addition, the impact of the expended incubation and diet change in the case of the EPN resistant individuals on the gut microbiota composition, leading to spreading of these samples in PCoA analysis, could not be excluded. As previously shown, the gut microbiota of laboratory-reared insects fed with the simple diet is different (usually less diverse) from that in wild living insects [43].”
L407-409: “Taken together, it is most probable that the changes observed in the midgut composition, apart from the extended incubation of larvae, are a result of the EPN treatment.”
L396: what about differences between nematode species, not really discussed in this section?
The issue of the impact of nematode species identity on the bacterial gut community has been elaborated in the Discussion subsection:
“Taking the impact of different EPN species exposure on the midgut microbiota community into account, we did not observe any significant differences. This is consistent with the findings reported by Cambon et al., who showed that the identity of three Steinernema spp. used for infestation of insect larvae had no effect on the composition of the gut bacterial community in the early step of the infection [43]. However, it is also possible that we were not able to detect the discrete changes due to the high intra-individual variability of microbiota compositions.
L442: advanced -> alternative
This has been corrected.
L450: independent -> dependent
This has been corrected.
